# Effects of Sodium Alginate and Calcium Chloride on Fungal Growth and Viability in Biomass-Fungi Composite Materials Used for 3D Printing

**DOI:** 10.3390/biomimetics9040251

**Published:** 2024-04-20

**Authors:** Al Mazedur Rahman, Caleb Oliver Bedsole, Yeasir Mohammad Akib, Jillian Hamilton, Taieba Tuba Rahman, Brian D. Shaw, Zhijian Pei

**Affiliations:** 1Department of Industrial & Systems Engineering, Texas A&M University, College Station, TX 77843, USA; almazedurrahman@tamu.edu (A.M.R.); yeasir.akib@tamu.edu (Y.M.A.); taieba_tuba@tamu.edu (T.T.R.); 2Department of Plant Pathology and Microbiology, Texas A&M University, College Station, TX 77845, USA; olib@tamu.edu (C.O.B.); jillianh2021@tamu.edu (J.H.)

**Keywords:** 3D printing, crosslinking, fungal growth, sodium alginate, biomass-fungi

## Abstract

To combat climate change, one approach is to manufacture products from biomass-fungi composite materials instead of petroleum-based plastics. These products can be used in packaging, furniture, and construction industries. A 3D printing-based manufacturing method was developed for these biomass-fungi composite materials, eliminating the need for molds, and enabling customized product design. However, previous studies on the 3D printing-based method showed significant shrinkage of printed samples. In this paper, an approach is proposed to reduce the shrinkage by incorporating ionic crosslinking into biomass-fungi composite materials. This paper reports two sets of experiments regarding the effects of sodium alginate (SA) and calcium chloride (CaCl_2_) on fungal growth and fungal viability. The first set of experiments was conducted using Petri dishes with fungi isolated from colonized biomass-fungi material and different concentrations of SA and CaCl_2_. Fungal growth was measured by the circumference of fungal colonies. The results showed that concentrations of SA and CaCl_2_ had significant effects on fungal growth and no fungal growth was observed on Petri dishes with 15% CaCl_2_. Some of these Petri dishes were also observed under confocal microscopy. The results confirmed the differences obtained by measuring the circumference of fungal colonies. The second set of experiments was conducted using Petri dishes with biomass-fungi mixtures that were treated with different concentrations of SA and exposure times in a CaCl_2_ (crosslinking) solution. Fungal viability was measured by counting colony-forming units. The results showed that the addition of the SA solution and exposure times in the crosslinking solution had statistically significant effects on fungal viability. The 2SA solution was prepared by dissolving 2 g of SA in 100 mL of water, the 5SA solution was prepared by dissolving 5 g of SA in 100 mL of water, and the crosslinking solution was prepared by dissolving 5 g of CaCl_2_ in 100 mL of water. The results also showed that fungal viability was not too low in biomass-fungi mixtures that included 2SA solution and were exposed to the crosslinking solution for 1 min.

## 1. Introduction

Significant challenges of our era encompass climate change and the depletion of natural resources. One way to address these challenges involves utilizing materials derived from renewable sources. One example is biomass-fungi composite materials.

Biomass-fungi composite materials primarily consist of two components: biomass particles sourced from agricultural waste (such as corn stover, beechwood sawdust, and hemp hurd) and a matrix of fungal hyphae that penetrates and connects the biomass particles [1]. These composite materials can be used for producing products that are typically made from petroleum-based plastics. Potential uses of these products span various industries, including packaging [2,3,4,5], furniture [6], and construction [7]. Biomass-fungi composite materials possess good thermal and acoustic insulation properties [8,9,10] and can biodegrade at the end of their service [11].

Utilizing 3D printing-based manufacturing methods [12] to produce products using biomass-fungi composite materials present an alternative to conventional molding-based manufacturing methods. Three-dimensional printing-based manufacturing methods enable the production of complexly shaped products in art, interior design, packaging, architecture, and construction [13,14,15,16], which are challenging to produce using molding-based manufacturing methods. The processes of one 3D printing-based manufacturing method are shown in Figure 1 and briefly described below.

The feedstock material for the first process is a biomass-fungi material. To prepare the biomass-fungi material, the biomass material is pasteurized at an elevated temperature to kill other microorganisms that might compete with the fungi for nutrition. Then, the fungi spores are added to the biomass material. This biomass-fungi material is dehydrated prior to being packed in sterilized filter patch bags.

Primary colonizing: water and wheat flour are added to the filter patch bag that contains the biomass-fungi material. The bag is kept in a sterilized environment for a certain number of days to allow fungi to grow on the biomass material to create a foam-like biomass-fungi composite.

Mixing: the foam-like biomass-fungi material prepared in primary colonizing is mixed with additives (such as water, psyllium husk, and wheat flour) for preparing the biomass-fungi mixture for 3D printing.

3D printing: samples are printed using the biomass-fungi mixture prepared by mixing.

Secondary colonizing: the printed samples are kept in a sterilized environment to allow further fungal growth in 3D printed samples.

Drying: the printed samples, after the secondary colonizing, are dried (for example, in a conventional oven) to kill all the fungi in these samples. If the fungi are not killed, they may restart growing when the printed samples are exposed to suitable humidity conditions.

There are reported studies [15,16,17,18,19] on 3D printing of biomass-fungi composite materials. In some of the reported studies [15,17,19], significant shrinkage was observed in printed parts during the secondary colonization process, causing poor shape fidelity of the printed parts. Shape fidelity is the ability of a printed part to maintain its shape compared with the shape in the computer design [20,21]. However, there are no reported studies focusing on increasing the shape fidelity of printed parts using biomass-fungi composite materials.

This paper reports one approach to improve the shape fidelity of printed parts by incorporating ionic crosslinking into biomass-fungi composite materials. Ionic crosslinking has been used in bioprinting with hydrogel-based bioinks [22,23]. In many reported studies [24,25,26] on bioprinting with hydrogel-based bioinks and ionic crosslinking, sodium alginate (SA) was used as the crosslinking agent and calcium chloride (CaCl_2_) as the crosslinking solution. SA is a natural polysaccharide comprising D-mannuronic (M) and L-gulunronic acids (G) and is commonly extracted from marine brown algae [27]. Effects of SA and CaCl_2_ on cell viability [28] have been studied with animal cells [29] and algae cells [28]. It has been shown that SA could provide essential nutrients and a suitable environment for fungal growth [26]. SA has also been used for the microencapsulation of fungi [30,31]. Additionally, SA can enhance the mechanical properties of 3D-printed products by making them more stable and durable [25]. The literature and the authors’ past experience indicate that a higher CaCl_2_ concentration could result in higher mechanical properties [32]. However, there are no reported studies regarding the effects of SA and CaCl_2_ on fungal growth and viability in biomass-fungi composite materials. This paper fills this gap by reporting the effects of SA and CaCl_2_ on fungal growth and viability in biomass-fungi composite materials.

## 2. Effects on Fungal Growth in Different Medium Solutions

Figure 2 shows the overview of the experimental procedure presented in this section.

### 2.1. Experimental Procedure

#### 2.1.1. Procurement of Materials

The biomass-fungi material was acquired from GROW.bio (Green Island, NY, USA) and was delivered in a polypropylene bag equipped with a square-shaped filter (1.5 × 1.5 inches in size) with a pore size of 0.2 µm. The biomass material comprised hemp hurd particles and the fungi strain was *Ganoderma lucidium* [19]. Based on the sieve analysis of the as-received material, more than 80% of the particles were greater than 1 mm in size. While 92% of the particles passed through a sieve with a mesh opening of 4.75 mm, only 4% of the particles passed through a sieve with a mesh opening of 600 µm. Wheat flour (all-purpose flour: Great Value, Walmart, Bentonville, AR, USA) and psyllium husk powder (NOW Supplements, Bloomingdale, IL, USA) were obtained from a Walmart store. Sodium alginate (SA) (with molecular weights of 12,000–40,000 daltons) powder was obtained from Milipore Sigma (Sigma-Aldrich, St. Louis, MO, USA). This SA powder was extracted from *Macrocystis pyrifera* and composed of approximately 61% mannuronic and 39% guluronic acids [33]. The ratio of M/G acids is approximately 1.56. Calcium chloride (CaCl_2_) granules were obtained from Honeywell Electronic Chemicals (Charlotte, NC, USA).

#### 2.1.2. Preparation of Primary Colonized Biomass-Fungi Material

A total of 32 g of wheat flour and 700 mL of water, after being autoclaved (120 °C for 30 min), were added to the filter patch bag containing the as-received biomass-fungi material. Afterward, the bag was vigorously shaken by hand for one minute. This bag was then stored in a closet at 23 °C for a period of five days to allow fungal growth (i.e., primary colonizing) [12].

#### 2.1.3. Preparation of HDPA Plates Containing Pure Fungi Culture

After the primary colonizing process, 1 g of the resulting primary colonized biomass-fungi material was put onto a plate containing half strength Potato Dextrose Agar (HPDA) [34] (19.5 g/L Difco PDA (Potato Dextrose Agar), adjusted to 1.5% agar) and allowed to grow for an additional five days. The growing mycelium was examined to confirm the presence of clamp connections, a characteristic feature of *Ganoderma lucidum* fungi [35,36], and subsequently isolated in HPDA media. Agar plugs were extracted from fully colonized HPDA plates containing the isolated fungus in pure culture [37].

#### 2.1.4. Preparation of Petri Dishes with Different Concentrations of SA and CaCl_2_

A total of 98 g of HPDA was mixed with 2 g of SA powder in a beaker. The nutrient-rich HPDA with chemicals was used as a growth medium solution. The growth medium solution was then autoclaved. Instantly after autoclaving, 20 mL of this growth medium solution was spread into sterile Petri dishes. This way, 2% SA Petri dishes were prepared.

The procedure to prepare the 5% SA Petri dishes was the same except that 5 g (instead of 2 g) of SA powder was mixed with 95 g of HDPA in a beaker. The procedure to prepare the 5% and 15% CaCl_2_ Petri dishes was the same except that 5 g of CaCl_2_ granules was mixed with 95 g HPDA and 15 g of CaCl_2_ was mixed with 85 g of HDPA, respectively. The control Petri dishes contained only 100 g of HDPA.

A total of 0.5 cm of agar plug inoculum (prepared by following the procedure described in Section 2.1.3) was placed in the middle of each of these Petri dishes.

#### 2.1.5. Evaluation of Fungal Growth Using Circumference of Fungal Colonies

In reported relevant studies, several evaluation methods for fungal growth were used, such as hyphal length, mycelium dry weight, and colony diameter or the circumference of fungal colonies [38]. In this study, the circumference of fungal colonies was used to measure fungal growth [38].

Table 1 shows the experiment matrix for measuring the effects of different concentrations of SA and CaCl_2_ on fungal growth. For each of the concentrations of SA and CaCl_2_, and control, four Petri dishes were prepared, with the total number of Petri dishes equaling 20.

After 0, 3, and 5 days, the circumference of the fungal colonies was measured under an Olympus SZX9 dissecting microscope (Olympus America, Center Valley, PA, USA) for each Petri dish sample. Pictures were taken after 0, 3, and 5 days of fungal growth for each Petri dish.

#### 2.1.6. Hyphae Growth Speed Observed Using Confocal Microscope

To confirm the results of fungal growth measured by the circumference of fungal colonies, hyphae were also imaged using a confocal microscope. From the fungal colonies in the Petri dishes grown, as described in Section 2.1.4, one single hypha was selected randomly from each Petri dish to measure the speed of hyphae growth. Confocal microscope observations were made using an Olympus FV3000 laser scanning confocal system affixed to an Olympus IX83 inverted microscope (Olympus America, Center Valley, PA, USA). This setup included a Galvanometer scanner and High Sensitivity GaAsP PMT detectors (Olympus America, Center Valley, PA, USA). The objective was the UAPON100XOTIRF (NA = 1.49) with a 405 nm laser with Differential Interference Contrast (DIC) microscopy. Imaging was conducted by using the agar block method [39]. Imaging over the course of 4 min 16 s and 8 min 33 s were captured.

### 2.2. Results and Discussion

Both *t*-test (one-tailed or two-tailed as necessary) and Tuckey pairwise comparison were conducted using Minitab software (version 2019) on the experiment results. The *p*-value represents the minimum significance level at which the difference is statistically significant [40,41]. In other words, if the significance level is set at a value below the *p*-value, the difference will not be statistically significant. In this context, the significance level of 0.05 is employed to determine the statistical significance of differences when presenting experimental results.

#### 2.2.1. Effects on Fungal Growth Measured by Circumference of Fungal Colonies

Table 2 shows pictures of Petri dishes prepared for different test conditions by following the procedure described in Section 2.1.4. The data on circumferences of the fungal colonies in these Petri dishes are presented in Figure 3. The error bars in the figure represent 95% confidence intervals of the means. The day 0 mean circumference was displayed as 0 cm in Table 2 for all the Petri dishes, and if there was no growth at 3 or 5 days, it was also indicated as 0 cm circumference. The control Petri dishes had the highest fungal growth (circumference = 23.34 cm). The 2% SA Petri dishes had significantly lower fungal growth (*p*-value = 0.004 < 0.05) than the control Petri dishes. The 5% SA Petri dishes also had significantly lower fungal growth (*p*-value = 0.0001 < 0.05) than the control Petri dishes. The 2% SA Petri dishes had significantly higher fungal growth (*p*-value = 0.0001 < 0.05) than the 5% SA Petri dishes. In summary, among the tested conditions, the addition of SA into Petri dishes reduced fungal growth. The results are consistent with the results reported by Jung et al. [42], who found that chitosan and SA multilayer coatings inhibit fungal growth. Tøndervik et al. [43] demonstrated the antifungal properties of an oligosaccharide, an alginate derived from seaweed.

The circumferences of fungi colonies for the 5% CaCl_2_ and 15% CaCl_2_ Petri dishes were 3.29 cm and 0.04 cm, respectively. The 5% CaCl_2_ and 15% CaCl_2_ Petri dishes had significantly lower fungal growth (*p*-value = 0.00001 < 0.05 and *p*-value = 0.00001 < 0.05, respectively) compared with the control Petri dishes. The 5% CaCl_2_ Petri dishes showed a very small amount of fungal growth (visible in the microscope) compared to the control Petri dishes, but using a higher concentration of CaCl_2_, more than 5%, would be more detrimental. The results agreed with the results obtained by Boumaaza et al. [44]. Boumaaza et al. studied the effects of CaCl_2_ on mycelium growth and found that higher concentrations of CaCl_2_ reduced fungal (mycelium) growth significantly. So, for further investigation, only 5 g CaCl_2_ was mixed with 100 mL water to prepare the crosslinking solution. The result further supports the results of Maouni et al. [45], who found that CaCl_2_ significantly reduced pear fruit decay caused by *A. alternata* and *Penicillium expansum* when used at 4 and 6%. Tian et al. [46] recorded that calcium chloride at 2% inhibited the growth and spore germination of *R. stolonifera*. Maintaining low calcium levels is imperative for filamentous fungi, as optimal calcium signaling and homeostasis play a pivotal role in facilitating hyphal growth, differentiation, and virulence [47]. Experiments on fungi have shown that mutants that have defective intracellular Ca^2+^ transport systems or defective vacuolar H^+^-ATPase that produces the proton motive force necessary for the activity of the vacuolar Ca^2+^/H^+^ exchanger [48] could not grow in high Ca^2+^ concentrations [49,50]. Maintenance of low basal concentrations of free cytosolic Ca^2+^, in the submicromolar range, is essential for normal cell functions [51,52].

#### 2.2.2. Effect on Fungal Growth Observed under the Confocal Microscope

Based on the results presented in Section 2.2.1, only Petri dishes of control, 2% SA, and 5% SA showed good fungal growth. These Petri dishes were used for confocal microscopy following Section 2.1.6. Images are shown in Figure 4. Control Petri dishes had a higher rate of hyphae growth than the 2% SA and 5% SA Petri dishes. These results confirm the trends obtained from Section 2.2.1.

## 3. Effects on Fungal Viability in Biomass-Fungi Mixture

Figure 5 shows the overview of the experimental procedure used in this section.

### 3.1. Experimental Procedure

#### 3.1.1. Procurement of Materials

Procurement of materials was described in Section 2.1.1.

#### 3.1.2. Preparation of Primary Colonized Material

Primary colonized biomass-fungi material was prepared following the procedure described in Section 2.1.2.

#### 3.1.3. Preparation of Sodium Alginate Solution

The preparation procedure of the SA solution with the concentration of 2:100 (2SA) (*w*/*v*) is illustrated in Figure 6 and described below [28,53].

Step 1: A 500-mL beaker with 100 mL of autoclaved water was put on a hot plate magnetic stirrer (Thermo Fisher, Waltham, MA, USA) and the temperature was set to 60 °C.

Step 2: A polytetrafluoroethylene (PTFE) coated magnetic stir bar was placed in the beaker and was set to rotate at 800 rpm.

Step 3: A total of 2 g of SA powder (Sigma-Aldrich, Saint Louis, MO, USA) was prepared using a Ohaus Sp-202 Scout Pro (OHAUS Corporation, Parsippany, NJ, USA) weight balance. The SA powder was slowly added into the beaker to prevent the SA powder from clumping. The beaker was stirred for 2 h.

The procedure to prepare the SA solution with a concentration of 5:100 (5SA) (*w*/*v*) was the same except for Step 3. Instead of 2 g, 5 g of SA powder was added into the beaker. The 0SA represented 100 mL of water (no SA was added).

#### 3.1.4. Preparation of Biomass-Fungi Mixtures with Different SA Concentrations

A total of 50 g of colonized biomass-fungi material, 200 mL of SA solution (either 0SA, 2SA, or 5SA), and 20 g of autoclaved wheat flour were added in the mixer cup of a mixer (NutriBullet PRO: Capital Brands, Los Angeles, CA, USA). Mixing parameters used were as follows: mixing time (the duration of mixing) = 15 s and mixing mode = intermittent (meaning that the mixing was performed for a duration of five seconds and stopped, and then, the mixer was shaken twice manually to ensure good contact of the mixture with the blades. Subsequently, the mixing resumed again). Thereafter, 10 g of psyllium husk powder was added to the mixer cup and mixed into this mixture using a spatula. This powder prevented the separation of phases in the biomass-fungi mixture while printing. In this way, a total of three biomass-fungi mixtures (labeled as 0SA, 2SA, and 5SA) with different SA concentrations were prepared.

#### 3.1.5. Preparation of Crosslinking Solution

The preparation procedure of the CaCl_2_ (crosslinking) solution with the concentration of 5:100 (*w*/*v*) [54,55] is illustrated in Figure 7 and described below [28].

Step 1: A 500-mL beaker was filled with 100 mL of autoclaved water and put on a hot plate magnetic stirrer (Thermo Fisher, USA).

Step 2: A PTFE-coated magnetic stir bar was placed in the beaker and was set to rotate at 800 rpm.

Step 3: A total of 5 g of CaCl_2_ granules (Sigma-Aldrich, Saint Louis, MO, USA) was measured using a weight balance (Ohaus Sp-202 Scout Pro, USA) and slowly added into the beaker. The stirring continued at room temperature for 30 min to ensure the total solvation of the CaCl_2_ granules in the water.

#### 3.1.6. Preparation of Samples for Measuring Fungal Viability in Biomass-Fungi Mixtures

The preparation was performed in the following steps.

Step 1: From each of the biomass-fungi mixtures (labeled as 0SA, 2SA, and 5SA), one block was created using a mold and a plunger (Figure 8).

Step 2: Then, the molded blocks were submerged in the crosslinking solution for different amounts of time (0 min for 0SA, and 1 min and 10 min for 2SA and 5SA).

Step 3: A sharp scalpel was first flame-sterilized using a gas burner until glowing red and was allowed to cool for 15 s. The spatula was used to gently streak a mass of one gram of biomass-fungi mixture from each block prepared in Step 2. Then, this mass was immediately suspended in a falcon tube containing 9 mL of autoclaved water.

Step 4: The content in the falcon tubes was then vortexed for 30 s. After vortexing, 100 microliters of the content were taken from the falcon tube and added to a centrifuge tube containing 900 microliters of water.

Step 5: A total of 100 microliters of the diluted solution in the centrifuge tube was then spread onto a Petri dish. Each Petri dish contained 20 mL of HPDA. After spreading the diluted solution on the HPDA Petri dish (plate), it was called the “Plated sample” in this study (Figure 8).

#### 3.1.7. Evaluation of Fungal Viability by Counting Colony Forming Units

Fungal viability (assessing viable fungal spores) in biomass-fungi mixtures was evaluated using colony-forming units per plate (CFUs/plate). This method was also used by other researchers in their reported studies [56].

Table 3 shows the experimental matrix for measuring the effects of different concentrations of SA and exposure times in the crosslinking solution on the fungal viability in the biomass-fungi mixture prepared for 3D printing. Three plated samples were prepared for each treatment. The experiments were replicated consistently over the course of three consecutive days. A total of seven treatments were prepared: 0SA without crosslinking exposure, 2SA with 0, 1, and 10 min exposure in the crosslinking solution, and 5SA with 0, 1, and 10 min exposure in the crosslinking solution. Consequently, a total of 63 experiments were conducted.

### 3.2. Results and Discussion

Both *t*-test (one-tailed or two-tailed as necessary) and Tuckey pairwise comparison were conducted using Minitab software on the experiment results. The *p*-value represents the lowest significance level at which the difference is statistically significant.

The results are presented in Figure 9. The error bars in the figure indicate 95% confidence intervals of the means. The 0SA plated samples had the highest fungal viability (colony forming units = 243 CFUs/plate) (Figure 9a). The 2SA without crosslinking plated samples had significantly lower fungal viability (*p*-value = 0.0001 < 0.05) than the 0SA plated samples (Figure 9a). The 5SA without crosslinking plated samples also had significantly lower fungal viability (*p*-value = 0.001 < 0.05) than the 0SA plated samples. However, there were no significant differences in fungal viability between 2SA without crosslinking plated samples and 5SA without crosslinking plated samples (*p*-value = 0.0001 < 0.05). In summary, among the tested treatments, the addition of SA into biomass-fungi mixtures reduced fungal viability, but the change in concentration of SA did not have a significant impact on fungal viability in biomass-fungi mixtures prepared for 3D printing.

Compared with both 2SA and 5SA without crosslinking plated samples, fungal viability in 2SA with 10 min crosslinking plated samples, 5SA with 1 min or 10 min crosslinking plated samples, was significantly lower (all their *p*-values were less than 0.05) (Figure 9b). All the plated samples with exposure to the crosslinking solution demonstrated a reduction in fungal viability except the 2SA with 1 min crosslinking plated samples. Fungal viability of 2SA with 1 min crosslinking plated samples was not significantly different from those of 2SA without crosslinking plated samples (*p*-value = 0.051 ≥ 0.05) and 5SA without crosslinking plated samples (*p*-value = 0.659 ≥ 0.05). So, the addition of 2SA and 1 min exposure crosslinking was the most favorable treatment for obtaining a high fungal viability among the tested SA concentrations and exposure times in crosslinking solution. Therefore, 2SA and 1 min crosslinking treatment can be used to prepare mixtures for 3D printing with biomass-fungi composite materials.

A recent study [57] showed that fungal viability had significant effects on the mechanical properties of biomass-fungi samples prepared using molding-based methods. Higher fungal viability tended to enhance mechanical properties by facilitating the formation of interconnections between biomass particles through increased hyphal growth. However, excessive fungal growth, leading to over-digestion of biomass substrates, could potentially degrade mechanical properties. Despite the reduction in fungal viability due to the addition of SA and exposure to crosslinking solutions, understanding how different concentrations of SA and exposure times in the crosslinking solution affect fungal growth and viability could enable researchers to identify optimal conditions. This knowledge could lead to the development of biomass-fungi composite materials with improved print quality and favorable mechanical properties.

## 4. Concluding Remarks

This paper reports the effects of sodium alginate (SA) and calcium chloride (CaCl_2)_ on fungal growth and viability for 3D printing of biomass-fungi composite materials. The main conclusions are as follows:Five different types of Petri dishes with different concentrations of SA and CaCl_2_ were prepared. The control Petri dishes had the highest fungal growth (circumference = 23.34 cm). The 2% and 5% SA Petri dishes had significantly reduced fungal growth compared with the control Petri dishes.Based on results from the effects of different concentrations of SA and CaCl_2_ on fungal growth, only control, 2%, and 5% SA Petri dishes were used for confocal microscopy observations. The results showed that control Petri dishes had a higher rate of hyphae growth than the 2% SA and 5% SA Petri dishes.In the set of experiments using plated samples, biomass-fungi mixtures were treated with different concentrations of SA and exposure times in the crosslinking solution. Fungal viability was measured by counting colony-forming units. Among the tested concentrations, the 0SA plated samples had the highest fungal viability, and the addition of SA into biomass-fungi mixtures reduced fungal viability, but a change in concentration of SA did not make any significant difference in fungal viability. The results also showed that 2SA with a 1 min crosslinking treatment can be used to prepare mixtures with biomass-fungi composite materials for 3D printing. Crosslinking might improve the print quality and the mechanical properties of the 3D printed parts using biomass-fungi composite materials.

Future studies will include the effects of SA and CaCl_2_ (crosslinking solution) on printability, rheology, and mechanical and chemical properties of biomass-fungi composite materials. Also, the mechanisms of how CaCl_2_ and SA affect fungal growth and viability, as well as how to mitigate or eliminate their unfavorable effects on fungal growth and viability by incorporating other substances, will be investigated. Future studies will also include the determination of the optimal concentrations of SA and CaCl_2_ for incorporation with biomass-fungi composite materials used for 3D printing.

## Figures and Tables

**Figure 1 biomimetics-09-00251-f001:**
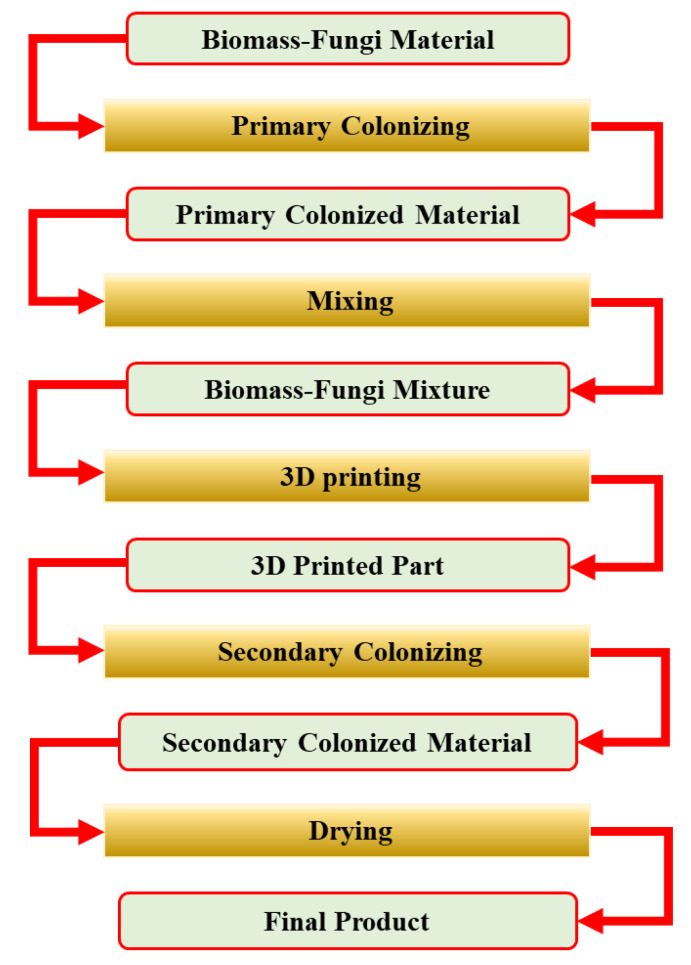
Processes of a 3D printing-based manufacturing method using biomass-fungi composite materials.

**Figure 2 biomimetics-09-00251-f002:**
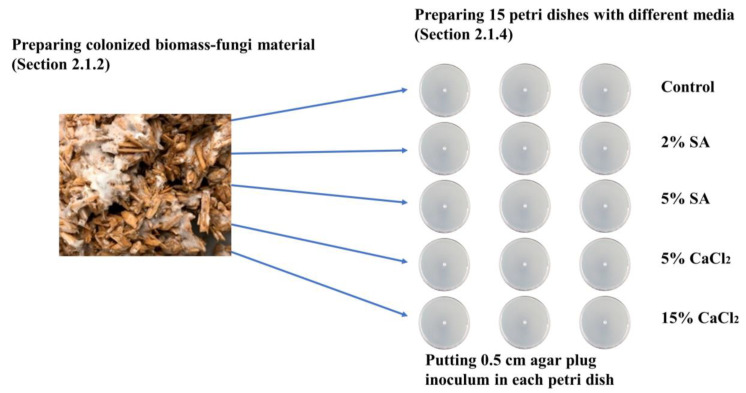
Overview of the experimental procedure for Section 2.

**Figure 3 biomimetics-09-00251-f003:**
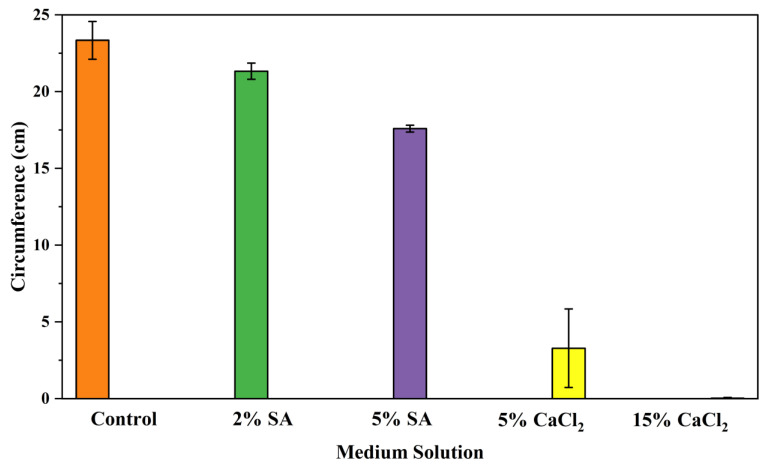
Mean values of the circumference of fungal colonies in Petri dishes with different medium solutions.

**Figure 4 biomimetics-09-00251-f004:**
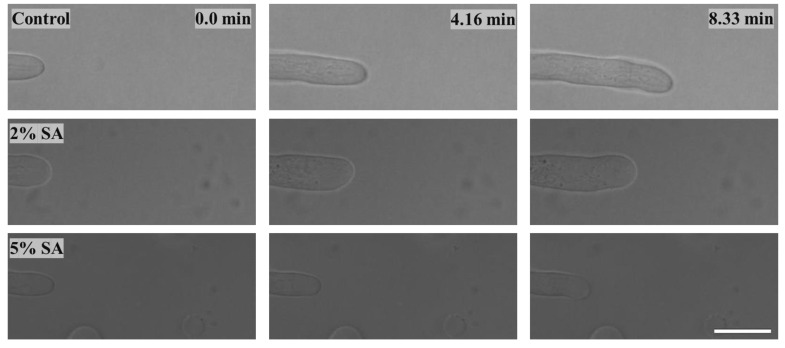
Timelapse images of hypha growth on Petri dishes with different medium solutions (control, 2% SA, and 5% SA), scale bar = 10 μm.

**Figure 5 biomimetics-09-00251-f005:**
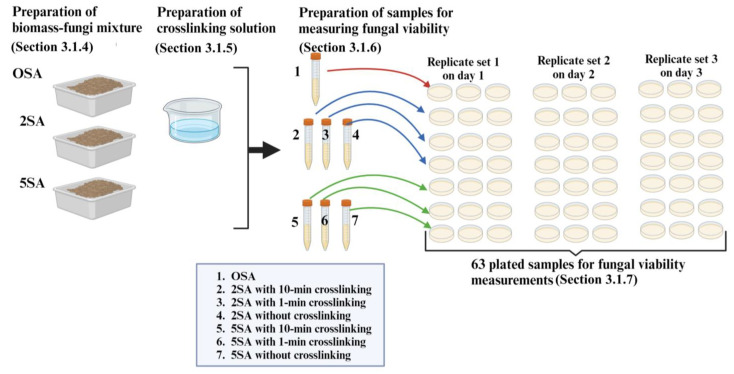
Overview of the experimental procedure used in Section 3.

**Figure 6 biomimetics-09-00251-f006:**
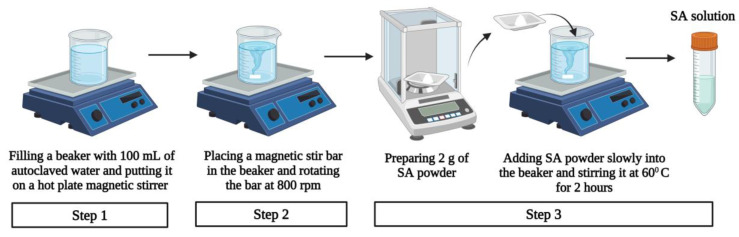
Preparation procedure of SA solution with the concentration of 2:100 (2SA) (*w*/*v*).

**Figure 7 biomimetics-09-00251-f007:**
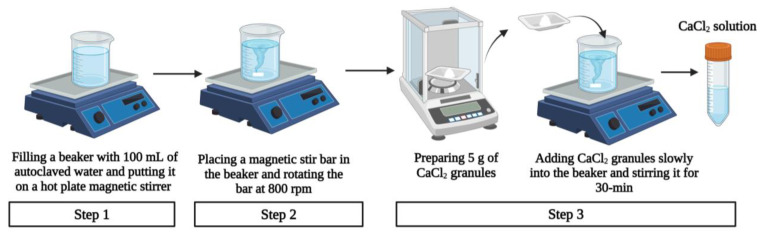
Preparation procedure of crosslinking solution with the concentration of 5:100 (*w*/*v*).

**Figure 8 biomimetics-09-00251-f008:**
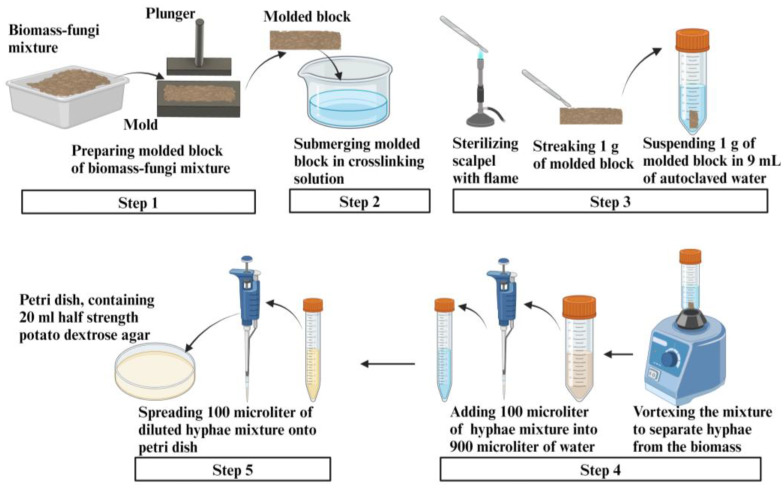
Preparing procedure for plated samples.

**Figure 9 biomimetics-09-00251-f009:**
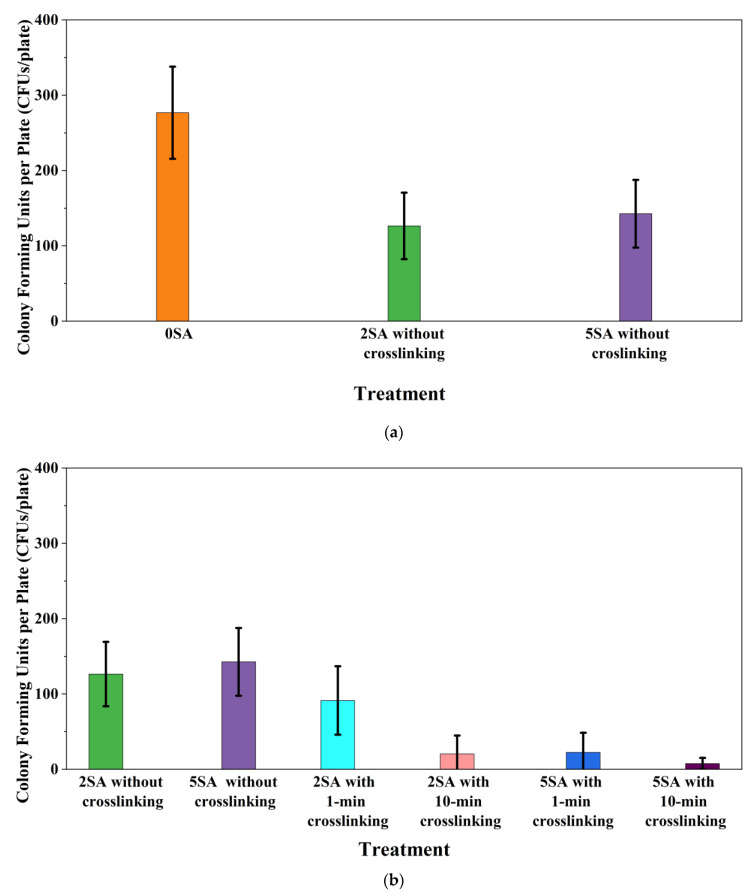
Colony Forming Units per plate (CFUs/plate) based on different treatments, (**a**) 0SA vs. 2SA and 5SA without crosslinking plated samples, (**b**) 2SA and 5SA without crosslinking vs. with crosslinking plated samples.

**Table 1 biomimetics-09-00251-t001:** Experiment matrix to study the effects of SA and CaCl_2_ on fungal growth.

	Concentration (%)	Number of Samples
Control (No SA or CaCl_2_)	0	4
SA	2	4
	5	4
CaCl_2_	5	4
	15	4

**Table 2 biomimetics-09-00251-t002:** Pictures of Petri dishes with different concentrations of SA and CaCl_2_ on Day 0, 3, and 5.

Petri Dish	Day 0	Day 3	Day 5
Control	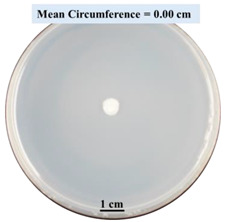	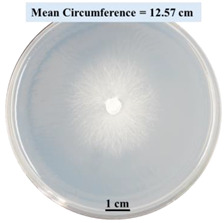	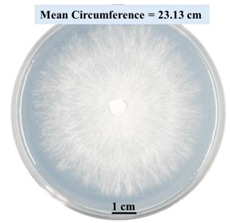
2% SA	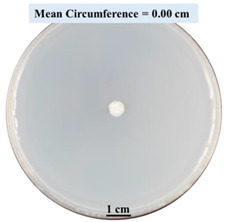	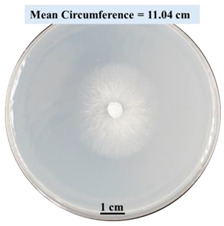	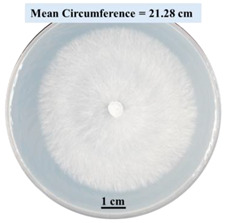
5% SA	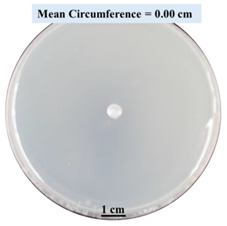	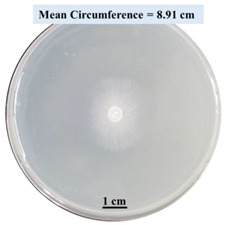	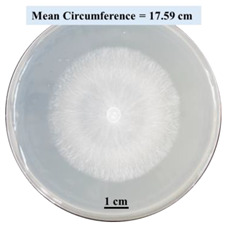
5% CaCl_2_	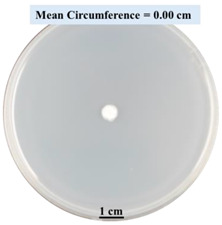	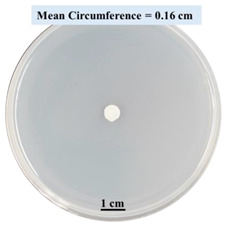	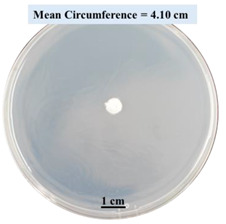
15% CaCl_2_	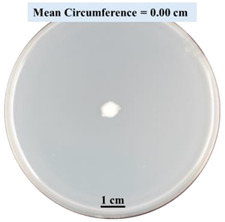	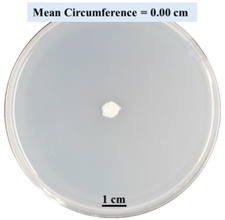	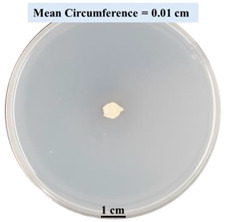

**Table 3 biomimetics-09-00251-t003:** Experiment matrix to study the effects of different concentrations of SA and exposure times in crosslinking solution on fungal viability in biomass-fungi mixtures prepared for 3D printing.

	Crosslinking Exposure Time (min)	Total Number of Plated Samples	Treatment
0SA	No crosslinking	9	0SA
2SA	0	9	2SA without crosslinking
	1	9	2SA with 1 min crosslinking
	10	9	2SA with 10 min crosslinking
5SA	0	9	5SA without crosslinking
	1	9	5SA with 1 min crosslinking
	10	9	5SA with 10 min crosslinking

## Data Availability

The authors confirm that the analyzed data to support the findings of this study are available within the article and others are upon request.

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
