# Peer review of "Effects of Sodium Alginate and Calcium Chloride on Fungal Growth and Viability in Biomass-Fungi Composite Materials Used for 3D Printing"

_biomimetics, 2024, doi:10.3390/biomimetics9040251_

Round 1
Reviewer 1 Report
Comments and Suggestions for Authors
In this effort, the authors have microencapsulated dry yeast in alginate and are characterizing growth. There are several claims in the abstract and the article that need to be removed for the material presented to be assessed. There are details missing in the methodology that need to be added for clarification of the chosen methods.
1) There is no evidence of 3D-printing nor a new method of fabrication. The microencapsulation of wet yeast in alginate has already been published.
2) There is no comparative wet growth curve in parallel with the dry yeast data to support some of the claims.
3) There are multiple references missing in the copy consulted.
4) What it the MW of alginate? What is the ratio of M/G acids? Medium MW alginate does not dissolve in 2 hrs and if you heat the solution the polymer undergoes hydrolysis. Please clarify.
5) Although there are graphs showing the circumference of growth, there is no size scale on the micrographs.
6) Why increase the CaCl2 concentration? The standard for bacterial and mammalian is 1.5% with no documented detrimental effects longterm.
Author Response
Howdy,
Thank you for the comments. The authors have revised the manuscript according to the reviewer's comments.
Regards,
Al Mazedur Rahman

Reviewer 2 Report
Comments and Suggestions for Authors This article showed the investigation of the influence of both sodium alginate and calcium chloride on the fungal growth and viability, the main component of a biomass. However the title of the manuscript is misleading for the readers. In detail, authors mentioned the "extrusion-based 3D printing of bio-3 mass-fungi biocomposite materials" (that would be the novelty of this investigation), but the relationship between reported data and the 3D printing properties of biomass-alginate mixture is not investigated. Therefore, a better fitting of the title, the purpose and the results should be presented. Moving to the body of the manuscript, all the results about biomass-alginate mixture biological properties are consistent, with an accurate statistical investigation, but there is a lack in the discussion of them in relation to the final application of samples. Moreover, the structure of the manuscript is complicated, with a continuous switch between methods (that are well explained by graphical representation) and results that negatively affect the ease of reading/understanding.Author Response
Howdy,
Thank you for the comments. The authors have revised the manuscript according to the reviewer's comments.
Regards,
Al Mazedur Rahman

Reviewer 3 Report
Comments and Suggestions for Authors
This present study explores the effect of sodium alginate and calcium chloride on Fungi growth. Fungi viability is evaluated using two indicators: fungi colony perimeter and colony formation per milliliter. In addition, statistical methods such as ANOVA and one-tail t-test are used to verify the significance of the above factors I believe this paper will be of interest to the readership of Biomimetics. However, a major revision is necessary before considering publication due to several questions raised and mistakes identified in this manuscript.
1. How do calcium chloride and sodium alginate affect the growth of fungi? Is it possible to mitigate or eliminate their affects by incorporating other substances?
2. The variable gradient is excessively large due to the limited number of experimental groups, which hinders its ability to accurately depict whether the condition of 2% sodium alginate (w/m), 1% calcium chloride (w/m), and a 1-minute CaCl2 exposure time represents the optimal condition.
3. When conducting statistical analysis, the researcher did not present the results or the procedures of the mathematical operations, thereby raising concerns about the credibility of the ANOVA or t-test outcomes. The t-value and confidence interval were also not provided.
4. The disparities in circumferences between 0% SA on Day 5 and 2% SA on Day 5, as indicated in Table 3, appear to be more pronounced compared to the differences between measurements of 7.52 cm and 7.36 cm. The results need to be carefully reviewed.
5. The colony forming units per plate of 5SA and no cross-linking, as shown in Figure 9, are greater than those of 2SA and no cross-linking, which contradicts the previous conclusion.
6. The whole story seems to have limited relevance to 3D printing. It would be more captivating if it further incorporates demonstrations of 3D printed fungi materials as reported in this study.
Comments on the Quality of English LanguageThe references errors and typos should be thoroughly revised.
Author Response

(The authors gave the same response as above.)

Round 2
Reviewer 1 Report
Comments and Suggestions for Authors
The researchers have addressed a sub-set of comments but not all. The successful fungi extraction and mixing with alginate resulting in growth are good scientific results however the major comment regarding the 3D printing method definition is not addressed.
1) The researchers have not addressed comment 1.
Specifically, the literature review has been revised with 3D printing literature relevant to fungi biocomposite extrusion, however there is still no evidence of 3D printing in this paper. The title, figures and any other claim to 3D printing in this work should be revised in the absence of this evidence.
3D printing requires specific physical and rheological measurements and variables none have been measured in the experiments other than observations regarding morphology. Crosslinking time and calcium chloride concentration do not constitute a complete set of 3D-printing parameters.
2) Visual quantitative imaging of growth still requires improvement (Table 3). A geometric variable or contrast imaging should be used aside from the ruler to substantiate statistical analysis.
Author Response

(The authors gave the same response as above.)

Reviewer 3 Report
Comments and Suggestions for Authors
The authors have effectively addressed some of my concerns, resulting in significant improvements to this paper. However, the experimental results exhibit significant disparities in the impact of different SA concentrations on fungal growth between the original manuscript and this revised edition (Figure 3). This paper canbe accepted after minor revision.
Comments on the Quality of English LanguageThe introduction is excessively long and requires further refinement.
Author Response

(The authors gave the same response as above.)
